# Fungal microbiome in gut of systemic lupus erythematosus (SLE)-prone mice (pristane and FCGRIIb deficiency), a possible impact of fungi in lupus

Thanya Cheibchalard[1], Asada Leelahavanichkul[2,3]*, Piraya Chatthanathon[4,5], Piriya Klankeo[6], Nattiya Hirankarn[2,7], Naraporn Somboonna[4,5,6]*

1 Program in Biotechnology, Faculty of Science, Chulalongkorn University, Bangkok, Thailand, 2 Department of Microbiology, Faculty of Medicine, Chulalongkorn University, Bangkok, Thailand, 3 Center of Excellence on Translational Research in Inflammation and Immunology (CETRII), Faculty of Medicine, Chulalongkorn University, Bangkok, Thailand, 4 Department of Microbiology, Faculty of Science, Chulalongkorn University, Bangkok, Thailand, 5 Multi-Omics for Functional Products in Food, Cosmetics and Animals Research Unit, Chulalongkorn University, Bangkok, Thailand, 6 Omics Sciences and Bioinformatics Center, Chulalongkorn University, Bangkok, Thailand, 7 Center of Excellence in Immunology and Immune-Mediated Diseases, Chulalongkorn University, Bangkok, Thailand

* Naraporn.S@chula.ac.th (NS); aleelahavanit@gmail.com (AL)

**Data Availability Statement:** Nucleic acid sequences in this study were deposited in an NCBI

## Abstract

The gut mycobiota (fungal microbiota) plays a crucial role in the immune system, potentially impacting autoimmune diseases such as systemic lupus erythematosus (SLE). Despite growing interest, data on intestinal fungi in SLE remain limited. This study thereby investigated the human-mimicked (mice) gut mycobiome and quantitative gut mycobiome analyses using universal fungal internal transcribed spacer 2 (ITS2) DNA next generation sequencing and real-time PCR, tracking time-series dynamics from preclinical to established SLE conditions in two SLE-prone mouse models. These models included pristane-induced mice, representing an environmental cause of SLE, and Fc gamma receptor RIIb (FcgRIIb) deficiency mice, representing a genetic factor. Fecal samples and different intestinal sections from mice aged 2–10 months were analyzed, including samples from 4-month-old and 11-month-old mice, which represented preclinical lupus (negative for anti-dsDNA) and established SLE conditions (positive for anti-dsDNA with proteinuria), respectively, alongside age-matched healthy controls. Results showed increased fungal diversity, specific changes in gut fungal species (i.e. increased *Candida* spp.), and an elevated Basidiomycota-to-Ascomycota (Basidiomycota/Ascomycota) ratio, which correlated with lupus activity in both lupus models. Linear discriminant analysis Effect Size (LEfSe; a possible representative organism) helped identify specific fungal difference between the lupus models. Our findings revealed that active lupus states may elevate gut fungal populations and alter fungal components in both the pristane and genetically susceptible SLE-prone mice, as indicated by mycobiota and quantitative mycobiota analyses. These changes could, in turn, influence disease activity. This research is essential for a deeper understand of the SLE-gut microbiome association, as the gut microbiome comprises both bacterial and fungal symbiosis. Manipulating fungal communities could present a potential therapeutic avenue for

open access Sequence Read Archive database, accession number PRJNA1106655.

**Funding:** This research was supported by the 90th Anniversary Chulalongkorn University Fund (Ratchadaphiseksomphot Endowment Fund, the Scholarship from the Graduate School of Chulalongkorn University to commemorate the 72nd Anniversary of his Majesty King Bhumibol Aduladej, and the Multi-Omics for Functional Products in Food, Cosmetics and Animals Research Unit. The funders had no role in study design, data collection and analysis, decision to publish, or preparation of the manuscript.

**Competing interests:** The authors have declared that no competing interests exist.

influencing disease outcomes in lupus. Further studies are crucial to clarify the direct role of gut fungi in lupus disease progression.

## Introduction

Systemic lupus erythematosus (SLE), or lupus, is a chronic autoimmune disorder characterized by systemic inflammation that affects multiple organs, including skin, joints, kidneys, brain, lungs, and blood vessels [1]. While various environmental, genetic, sex (more common in female), and hormonal factors contribute to its pathogenesis, the precise mechanisms remain unclear. Current immunosuppressive drugs help control lupus symptoms, but some serious side effects, particularly increased susceptibility to infections, can be fatal [2]. Therefore, a deeper understanding of the disease and the development of novel treatment strategies for SLE are essential.

Recent research links bacterial dysbiosis in the gut microbiota (an imbalance in bacterial diversity and relative abundances within the intestine) with SLE pathogenesis [3]. The gut microbiota includes various microorganisms, including bacteria, fungi and viruses, that inhabit the gastrointestinal (GI) tract and differ across regions of the GI tract due to variations in abiotic factors (e.g., pH, oxygen levels, and enzymes) and biotic factors (e.g., genetics). These microorganisms play a critical role in maintaining health [4–6]. The fungal community, as an important component of the gut microbiota, has also been associated with diseases like asthma, multiple sclerosis, and inflammatory bowel disease (IBD) [7–10]. Notably, enterocyte integrity (gut permeability) is essential as a barrier that separates microbial molecules from the host's bloodstream. In lupus, immune complex deposition in the intestine and intestinal vessels may impair gut permeability, microbial molecules to translocate from the gut into the bloodstream (a phenomenon known as "leaky gut"). This effect is evidenced by elevated beta-glucan (a fungal molecule) in the blood (glucanemia) of pristane-induced lupus and Fc gamma receptor IIb deficient (FcgRIIb-/-) mice [11, 12]. Pristane (2,6,10,14-tetramethylpenta-decane), a hydrocarbon derived from shark liver oil, induces lupus through chronic peritoneal inflammation, beginning as early as 20–24 weeks (wks) post-injection. Similarly, FcγRIIb deficiency (an inhibitory surface molecule on several immune cell, except for the T cells) causes spontaneous lupus onset in mice around 24 wks of age [11, 12]. Notably, FcγRIIb-/- mice develop anti-dsDNA antibodies, an autoantibody against self-DNA, as early as 2 months of age without other lupus characteristics, making them a useful preclinical lupus model. At 6 months of age, FcγRIIb-/- mice exhibit full-blown lupus with positive anti-dsDNA and lupus nephritis (proteinuria) [11–15]. Considering that (i) genetic variations may exist in lupus with similar clinical manifestations, (ii) human gut microbiome composition is influenced by genetics, age, gender, diet, environment medications, and GI conditions [16], (iii) FcγRIIb-/- polymorphisms are prevalent in Asian populations with lupus [17], and (iv) data on gut myco-biota from pristane-induced lupus mice (potentially representing environment-induced lupus) are lacking [18], the exploration of the gut microbiota in lupus models induced by FcγRIIb-/- deficiency and pristane injection (representing genetic and environmental lupus models, respectively) is crucial.

As glucanemia in lupus models were reported caused by either gut permeability defect or increased abundance of fungi in the gut [7–12, 19], gut mycobiota (from feces and different sections of the intestine) during SLE progression in SLE-prone mice were investigated in pristane and FcgRIIb-/- models, and age-matched healthy mice at 2–10 months old (from pre-

clinical SLE to established full-blown SLE lupus conditions). The alteration of gut fungi during lupus might be associated with lupus pathogenesis and might lead to a new therapeutic strategy; for example, a balance of bacteria and fungi in the gut.

## Materials and methods

### Animals, animal models, and clinical tests

The study protocol (protocol number 022/2561) was approved by the Institutional Animal Care and Use Committee of the Faculty of Medicine, Chulalongkorn University, Bangkok, Thailand (approval number 040/2561), following the National Institutes of Health, USA. Female mice were used in all experiments and FcGRIIb−/− mice (C57BL/6 background) were kindly provided by Dr. Silvia Bolland (NIAID, NIH, Maryland, USA), while the wild-type C57BL/6 mice were purchased from Nomura Siam International (Pathumwan, Bangkok, Thailand). Mice were housed in standard clear plastic cages (5 mice per cage) with free access to water and food in a light/dark cycle of 12-to-12 h in $22 \pm 2°C$ with $50 \pm 10\%$ relative humidity [12]. Because lupus characteristics demonstrated as early as 20 wks post-pristane injection and in 24-wk-old FcGRIIb−/− mice [13–15], pristane (Sigma-Aldrich, St. Louis, MO, USA) (0.5 ml) was administered in 4-wk-old WT mice to induce lupus approximately at 24-wk-old to match with FcGRIIb−/− mice. Because of age-dependent lupus characteristics, mice at 8- and 24-wk-old were used as representatives of asymptomatic and symptomatic lupus, respectively. For lupus characteristics, anti-dsDNA, urine protein, and serum cytokines (TNF-α, IL-6, and IL-10) were performed according to previous publications [11, 12]. Briefly, for anti-dsDNA, the calf DNA (Invitrogen, Carlsbad, CA, USA) was coated overnight in 96-well plates before added mouse serum and incubated with peroxidase-conjugated goat anti-mouse antibodies (BioLegend, San Diego, CA, USA), developed with ABTS peroxidase substrate solution (TMB Substrate Set; BioLegend), and read with a microplate photometer at a wavelength of 450 nm. Spot urine protein creatinine index (UPCI) was used to represent proteinuria as calculated: UPCI = urine protein (mg/dL)/urine creatinine (mg/dL), while urine protein and creatinine were measured by Bradford protein assay and QuantiChrom Creatinine-Assay, respectively. Serum cytokines were measured by ELISA (ReproTech, Oldwick, NJ, USA). At the completion of the study, mice were euthanized under isoflurane anesthesia to alleviate suffering.

### Sample collections and metagenomic extraction

The female C56BL/6 mice of each group, including wild-type (WT, or healthy), pristane induction (PT), and FcgRIIb knockout (KO) were used. Due to the mouse coprophagy habit (the eating of feces from other mice in same cage) of mice and the possible similar gut microbiome of mice in the same cage [11, 12], 3 mice were housed per 1 cage for total 3 cages per experimental group (WT, PT, and KO) and the fecal sample were collected from a representative mouse of each cage (the samples were collected only from these mice). Then, feces were sampled at 5 time points: 2, 4, and 6 (the disease onset of lupus), together with 8 and 10 months (full-blown lupus) of age. Additionally, five parts of the intestine including duodenum, jejunum, ileum, cecum, and colon were sampled at 4 months old as a representative of the preclinical SLE and at 11 months old as a representative of an established full-blown SLE. All mice were used to measure anti-dsDNA, UPCI, and serum cytokines IL-6, TNF-α and IL-10 to confirm the established SLE. During the established SLE phase, the PT and KO mice exhibited statistically higher anti-dsDNA antibody, pro-inflammatory cytokines TNF-α and IL-6, and proteinuria, than the WT mice, given that the overrepresentation of the immune responses were associated with lupus immunological disorders [20–22]. Notably, duodenum and colon were identified as the intestines at the distal part of stomach and cecum, respectively, while

ileum was the proximal part of the cecum and jejunum was 10 cm from stomach. Each sample (0.25 g) was extracted for metagenomic DNA using a DNeasy PowerSoil Kit, following the manufacturer's instructions (Qiagen, Maryland, USA). The quantity and quality of the extracted metagenomic DNA were analyzed by nanodrop spectrophotometry (A260 and A260/A280, respectively) and stored at -20˚C.

### ITS library preparation and MiSeq sequencing

The metagenomic DNA were analyzed for mycobiota by universal fungi DNA sequencing. First, a library preparation for the universal fungi region of internal transcribed spacer 2 (ITS2) was amplified via polymerase chain reaction (PCR) amplification using universal primers ITS3-KYO2 (5′-GATGAAGAACGYAGYRAA) and ITS4 (TCCTCCGCTTATTGATATGC) [23]. Each primer included an Illumina adaptor and barcode sequences as previously reported [19, 24–27]. The PCR cycling conditions were 94˚C for 3 min, 35 cycles of 94˚C 30 s, 52˚C for 30 s, 68˚C 45 s, followed by 68˚C 10 min. PCR amplicons were checked by 1.75% agarose gel electrophoresis. A minimum of three independent replicates were performed PCR to prevent bias and were purified with the GF-1 AmbiClean kit (Gel & PCR) (Vivantis, Selangor, Malaysia). Purified barcoded amplicons were measured concentrations using Qubit 3.0 Fluorometer and Qubit dsDNA HS Assay kit (Invitrogen, Waltham, USA) for 150 ng each. Each sample was pooled for MiSeq sequencing platform (Illumina, California, USA), along with the sequencing primers and index sequences. Sequencing was performed at the Omics Sciences and Bioinformatics Center, Chulalongkorn University (Bangkok, Thailand).

### Quantification of total fungi copy number

Quantification of total fungi in copy unit was determined via real-time PCR using universal ITS3_KYO2 and ITS4 primers in triplicates per reaction. Each 15 μL reaction mixture contained 7.5 μL of PerfeCta SYBR Green FastMix (Quantabio, Massachusetts, USA), 250 nM of each primer and 1 ng of metagenomic DNA, in a 48-well plate PCRmax Eco 48 Real-Time qPCR system (Cole-Parmer, Illinois, USA). The thermocycling parameters were 95˚C 5 min, followed by 40 cycles of 95˚C 15 s, 60˚C 30 s and 72˚C 30 s. Finally, a 60–95˚C melting curve analysis was performed to validate a single proper amplicon peak (i.e., neither primer-dimer nor non-specific amplification). Ten-fold serial dilutions of the ITS3_KYO2-ITS4 plasmids ($10^4$–$10^8$ copies/μL) were used as the reference standard curves in the fungal copy number computation as following equation [28–32].

$$\text{Copy number per } \mu L = \frac{\text{concentration (ng/}\mu\text{L}) \times 6.023 \times 10^{23} (\text{copies/mol})}{\text{length (bp)} \times 6.6 \times 10^{11} (\text{ng/mol})}$$

### Bioinformatics and statistical analyses for mycobiota and quantitative mycobiota diversity

Raw sequences (reads) were analysed following Mothur 1.39.5's standard operation procedures (SOP) for MiSeq [33]. Initially, reads were removed (a) reads shorter than 100 nucleotides (nt) excluding primer and barcode sequences, (b) ambiguous bases, and (c) chimera sequences. The quality sequences were pre-classified taxonomically with the UNITE ITS 8.3 database [34], and the non-fungal sequences (including protozoa, chromista, animals and plants) were removed. Then, the resulting quality sequences were clustered by phylotypes into phylum, order, class, family, genus, and species. Samples were normalized for an equal sequencing depth (3,421 quality sequences per sample). Alpha diversity including rarefaction curves, Good's coverage, operational taxonomic units (OTUs), Chao richness, and Shannon

diversity; and beta diversity including Bray-Curtis, non-metric multidimensional scaling (NMDS), and linear discriminant analysis effect size (LEfSe), were computed using Mothur 1.39.5 [32, 33]. The count of total fungi copy number from the ITS real-time PCR data were analyzed along with the percent mycobiota composition to yield the quantitative mycobiota (the fungal copy number for each individual OTU) [32, 35]. For general statistics, one-way ANOVA or Tukey's multiple comparisons test were used and a $P$-value $< 0.05$ was considered significant. For NMDS, analysis of molecular variance (AMOVA) with $P$-value $< 0.05$ was used. Graph pictures were prepared by GraphPad Prism 9.0 (San Diego, California, USA)

## Results

### Gut mycobiota alpha diversity and compositions in time-series healthy mice and SLE progression

The ITS library preparation and sequencing yielded 1,083,364 total raw reads and after quality sequence screening (which included removal of chimera and non-fungi sequences) yielded 947,423 total quality reads. These allowed $> 99\%$ Good's coverage (percent sequence coverage to true estimate) for all samples (S1 Table), which were consistent with the rarefaction curves that started reaching plateau (S1 Fig). The OTUs and Chao richness showed average slightly higher diversity in KO than PT and WT, respectively (S1 Table: OTUs 24 ± 2.71 WT, 28 ± 3.24 PT, 34 ± 4.30 KO; and Choa 32 ± 4.33 WT, 36 ± 4.61 PT, 45 ± 8.12 KO). This was consistent with the Shannon diversity (0.92 ± 0.08 WT, 1.20 ± 0.04 PT, 1.24 ± 0.09 KO). The KO group demonstrated a statistical increase in genus OTUs and Chao richness from 4 to 8 months of age (Fig 1A).

The gut mycobiota were dominated by phyla Ascomycota and Basidiomycota, and increased Basidiomycota was demonstrated as mice aged (after ~6–8 months) with the relatively greater Basidiomycota in PT and KO than WT (Fig 1B). The most abundance genera were *Saccharomyces* followed by *Candida*, and *Aspergillus* (Fig 1C). Comparing with the WT, the fungal composition patterns of SLE mice began to change at 2 months of age (before the onset SLE characteristics at 6 months) with the similar profiles between PT (lupus from external chemical stimulation) and KO (genetic factor) (Fig 1C). In lupus mice, *Saccharomyces* proportions reduced, while *Candida*, *Aspergillus*, and *Issatchenkia* abundance were expanded and the KO mice demonstrated higher *Blumeria* and *Nigrospora* than PT mice (Fig 1C).

### Quantification of mycobiota in feces of healthy mice and SLE progression

Following the quantification of fungi by the universal ITS real-time PCR, the number of fungal cell counts and the quantitative mycobiota compositions could be analyzed [32, 35]. Compared to age-matched WT, the total fungi count of both SLE-prone models showed a slightly increase at 6 and 8 months old but significantly decreased at 10 months old (Fig 2A). For quantitative analysis at phylum level, healthy or WT mice showed a steady rise in Ascomycota at 2 to 10 months (highest at 10 months), while highest at 6 months in both SLE-prone mice and became statistically lower than healthy mice at 10 months (Fig 2B).

At the species level, *Saccharomyces cerevisiae*, a possible beneficial organism [36] was statistically higher in WT than other groups at 10 months, while *Candida albican* in PT mice was higher than other groups at 2 months old after decreased into the closer level with WT and higher again finally at 10 months old (Fig 2C). Also, *C. albicans* in KO was highest among all groups at 10 months old (Fig 2C). In addition, both *Candida tropicalis* and *Issatchenkia orientalis* (*Candida krusei*) in lupus mice (PT and KO) were statistically higher than WT as early as 6 months old (a starting point of lupus nephritis in both lupus models [11, 12]), while *C.*

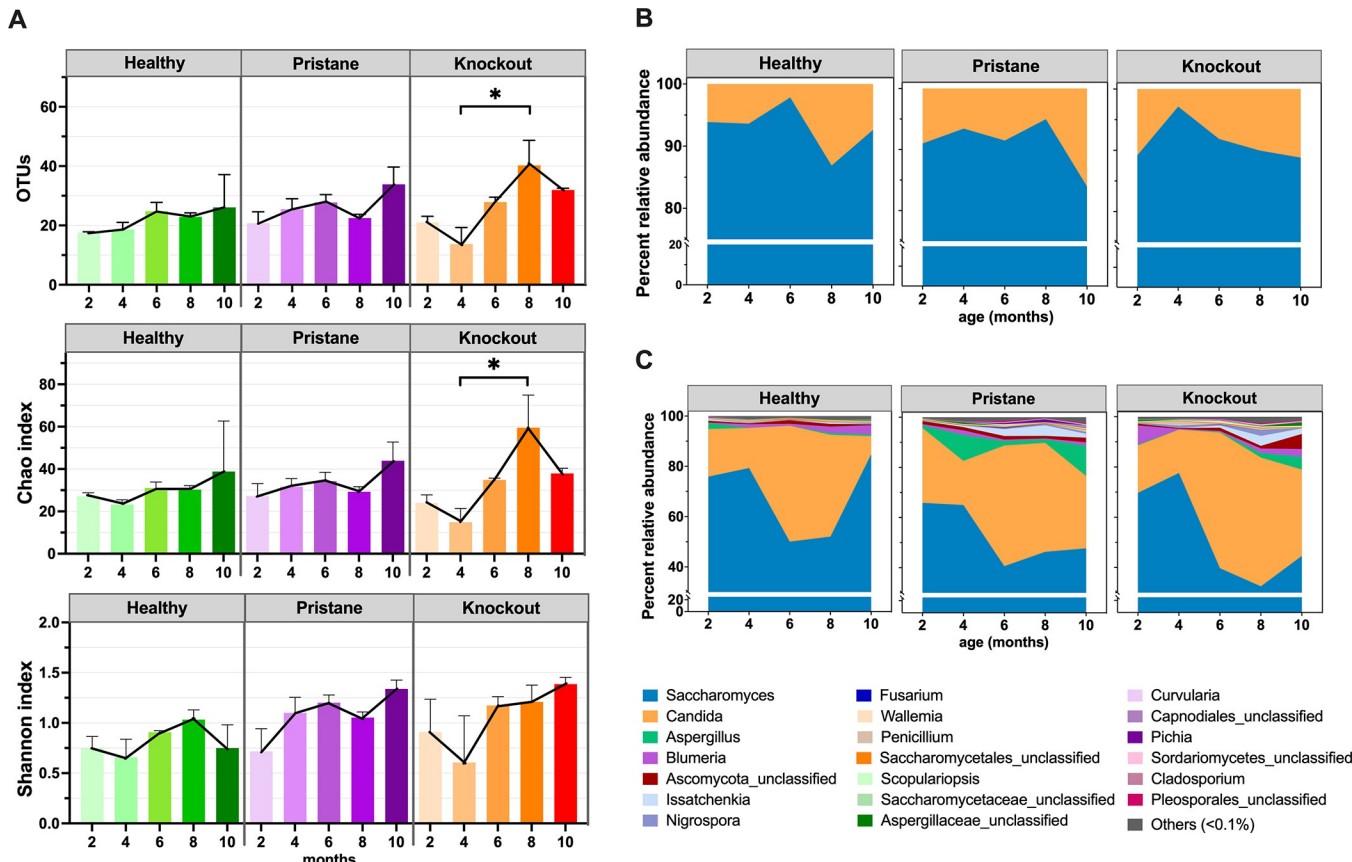

**Fig 1.** Comparison of (A) alpha diversity OTUs, Chao richness and Shannon diversity at genus level; and (B and C) relative time-dependent percent compositions of fecal mycobiota in phylum and genus OTUs. * represents statistical ANOVA $P < 0.05$.

*tropicalis* in KO was higher than PT at 8 months old and *I. orientalis* was lower than PT at 6 months old (Fig 2C). Meanwhile, *Aspergillus* increased in pristane-induced mice only at 4 months, and *Aspergillus* in knockout mice slightly and significantly increased at 6 and 10 months, respectively (S2 Fig).

The beta diversity (similarity or dissimilarity among samples) using non-metric multidimensional scaling (NMDS) revealed no cluster among groups (WT, PT, and KO) at 2–4 months, while the clustered data of WT and KO, but not PT were separated, at 6–8 months and significant separation between WT and SLE-prone mice were shown at 10 months old (Fig 2D). Consequently, we defined gut mycobiota at 4 months of age to represent a preclinical SLE state and gut mycobiota at 10–11 months of age as an established SLE state.

## Gut mycobiota compositions in different intestinal parts of healthy and SLE mice

The mycobiota diversity of young healthy mice (4 months) steadily increased from duodenum to colon and quite similar to old healthy mice (11 months). When compared between WT and SLE mice (PT and KO), the small intestines of both preclinical and established state were similar and in the large intestines of preclinical SLE not different (Fig 3A). In the large intestines at established state in PT mice showed significant increases of alpha diversity Chao and Shannon indices in cecum and colon, while KO mice showed a significant increase of OTUs, Chao and

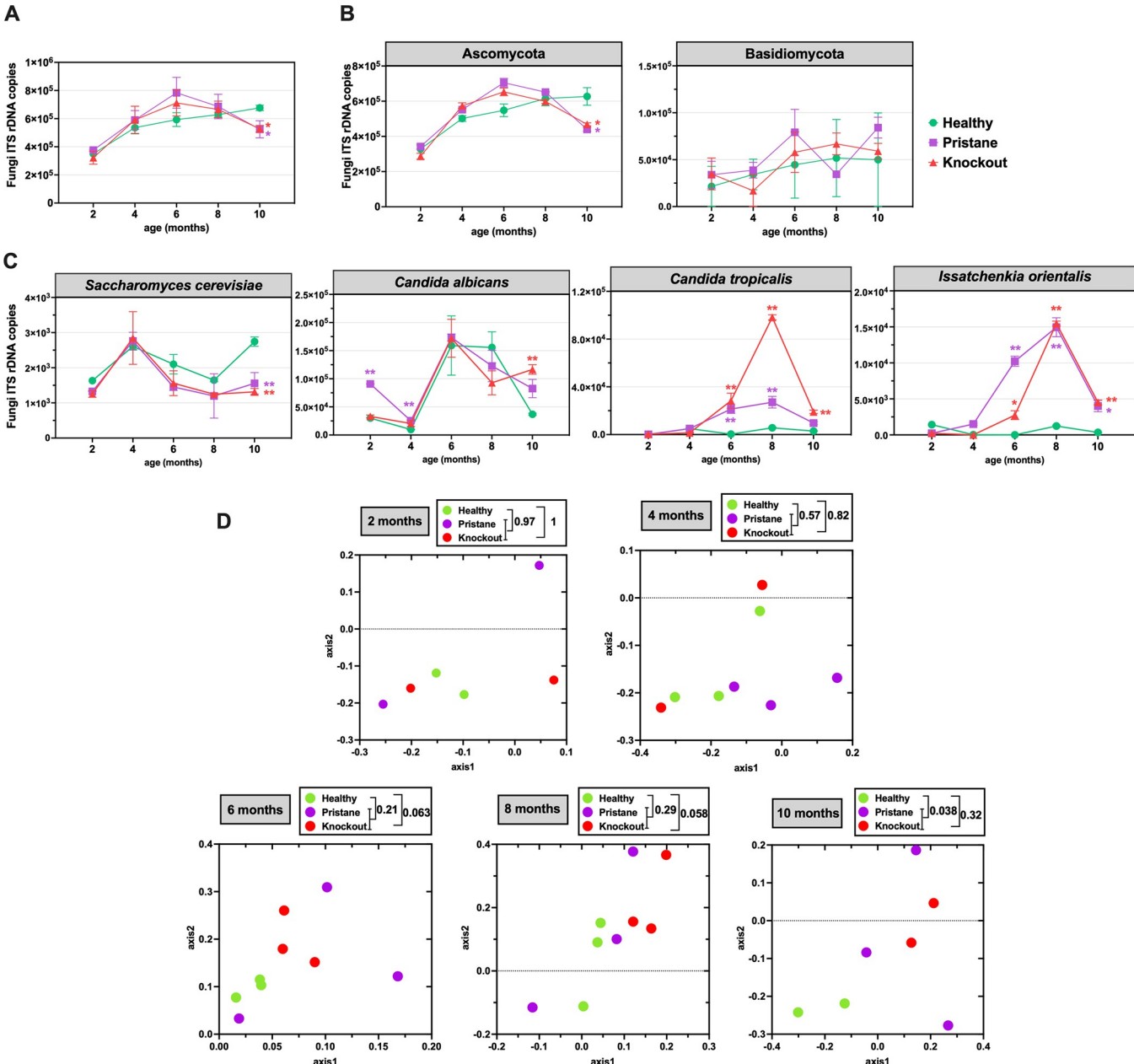

**Fig 2.** Comparison of relative time-dependent (A) quantitative fungal cell count, (B and C) counts of phylum and predominated statistically different species, and (D) beta diversity analysis via non-metric multidimensional scaling (NMDS), of fecal mycobiota. In (A-C), * and ** represent ANOVA $P < 0.05$ and $P < 0.01$, respectively. In (D), AMOVA was computed to test statistical difference ($P < 0.05$).

Shannon indices in cecum and colon (Fig 3A). These data suggested that established SLE mice demonstrated the increases of alpha diversity in large intestine in comparison with young healthy, old healthy and preclinical SLE.

In phylum level, fungi in each intestinal part were dominated by two phyla including Ascomycota and Basidiomycota in different proportions that were similar between WT and preclinical lupus mice (PT and KO) at 4 months old (Fig 3B). At 11 months, the patterns of fungal compositions were slightly different between WT and the established SLE mice as increased Basidiomycota in ileum, cecum, and colon of lupus mice (Fig 3B). Moreover, Basidiomycota

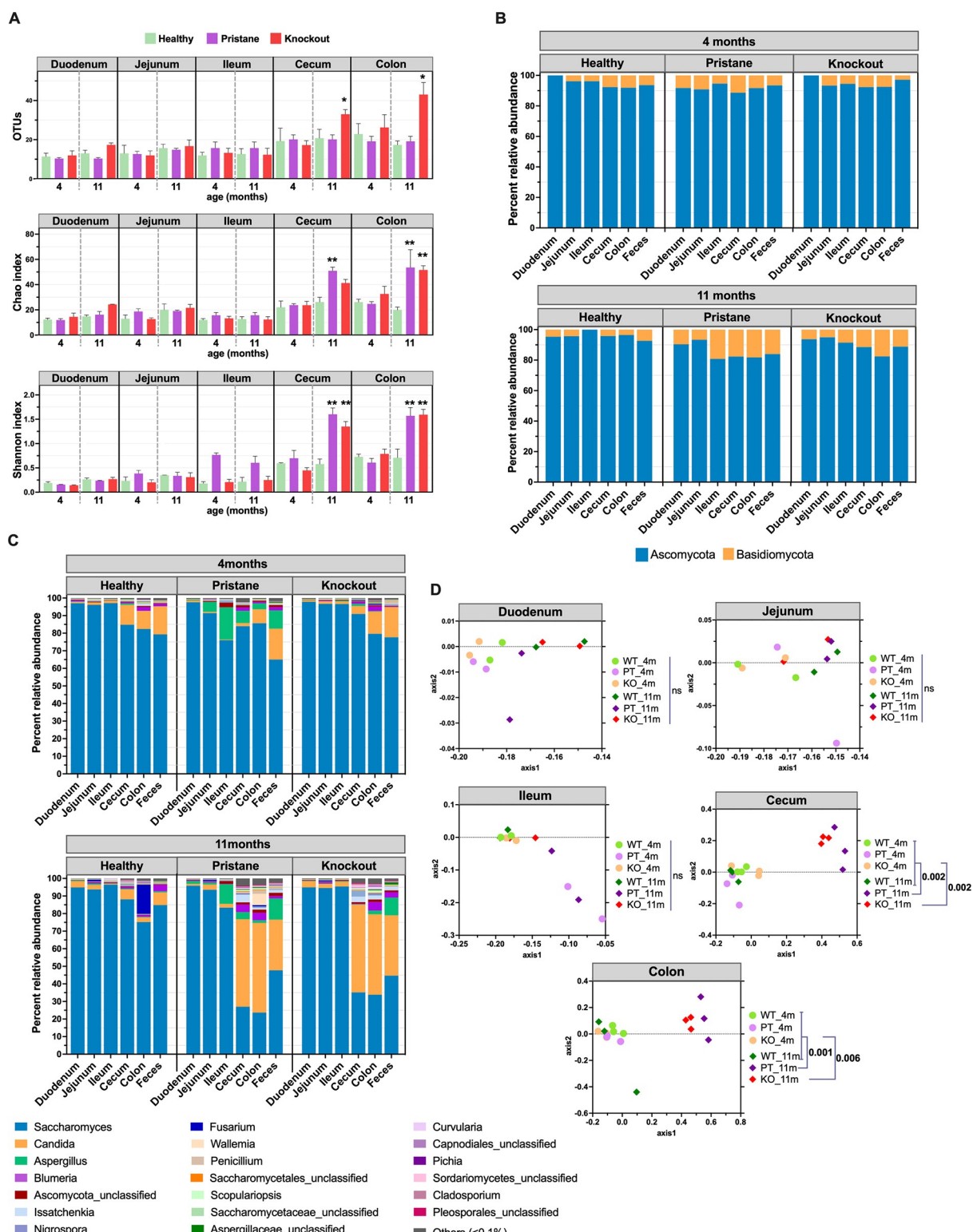

**Fig 3.** Different intestinal part mycobiota comparisons through (A) alpha diversity OTUs, Chao richness and Shannon diversity at genus level; (B and C) time-dependent percent compositions of fecal mycobiota in phylum and genus OTUs; and (D) beta diversity analysis NMDS along AMOVA statistics, among WT (or healthy), PT and KO mice. In (A), green color represents WT; purple, PT; and red, KO.

and Ascomycota ratio in these intestinal segments of established SLE mice (PT and KO) were significantly different from 11 months old WT (S3 Fig).

At genus level, the most abundant intestinal fungi were mostly *Saccharomyces*, followed by *Candida*, and *Aspergillus* (Fig 3C). The fungal composition patterns of WT at 4 months were alike to KO mice, while PT showed an increased *Aspergillus* (Fig 3C). In comparison with WT at 11 months, the established SLE mice (PT and KO) demonstrated obviously different mycobiota patterns (increasing *Candida*, *Aspergillus* and *Issatchenkia* in the large intestine) (Fig 3C). Notably, gut mycobiota compositions in phylum and genus level of cecum, colon and feces were mostly similar between PT and KO mice (Fig 3C).

## Beta diversity of mycobiota in different intestinal parts of healthy mice and SLE mice

Following the beta diversity NMDS and statistical analyses, there was non-different mycobiota in small and large intestines in asymptomatic mice, including WT (all age groups) and preclinical SLE, whereas mycobiota in the large intestines (caecum and colon) of the established lupus (6–11 months of PT and KO) were clearly separated from asymptomatic group (Fig 3D and Table 1). The mycobiota profiles of asymptomatic group and established SLE in cecum and colon were presented in comparative statistics in Table 1. However, we did not find any significant differences of gut mycobiota in cecum, colon and feces between PT and KO ($P > 0.05$) (Fig 3D).

## Quantification of mycobiota in the large intestine of healthy mice and SLE mice

Due to the altered mycobiota in the large intestines in lupus mice, quantitative mycobiota were also analyzed. While total fungal quantity among WT and preclinical SLE was not different, higher fungal counts in established PT and KO in the colon alone and both cecum and colon, respectively (Fig 4A), with predominant ratios of phyla Basidiomycota-to-Ascomycota (B/A ratio) in caecum and colon (Fig 4B). This B/A ratio in the large intestines of established SLE mice were significantly higher than age-match healthy mice (Fig 4B). In comparison of established SLE mice with each group at 4 months and healthy mice at 11 months, both PT and KO demonstrated decreased *Saccharomyces* and an increase in *Candida*, *Issatchenkia*, *Aspergillus*, *Wallemia*, *Penicillium* and *Cladosporium* in either cecum or colon (Fig 4C and S4 Fig).

**Table 1. AMOVA statistics of gut mycobiota compositions at genus OTUs between sample groups of healthy mice at 4 months (young) and 11 months (old), and SLE mice (combined PT and KO) at 4 months (preclinical SLE) and 11 months (established SLE) ($P < 0.05$).**

| Gut sections | | Groups | | p-value |
|---|---|---|---|---|
| **Large intestine** | **Cecum** | Young healthy | Old healthy | 0.3 |
| | | Young healthy | Preclinical SLE | 0.094 |
| | | Preclinical SLE | Established SLE | 0.089 |
| | | Healthy and preclinical SLE | Established SLE | 0.003* |
| | **Colon** | Young healthy | Old healthy | 0.118 |
| | | Young healthy | Preclinical SLE | 0.389 |
| | | Preclinical SLE | Established SLE | 0.007* |
| | | Healthy and preclinical SLE | Established SLE | 0.004* |
| **Feces** | | Young healthy | Old healthy | 0.107 |
| | | Young healthy | Preclinical SLE | 0.591 |
| | | Preclinical SLE | Established SLE | 0.1 |
| | | Healthy and preclinical SLE | Established SLE | 0.052 |

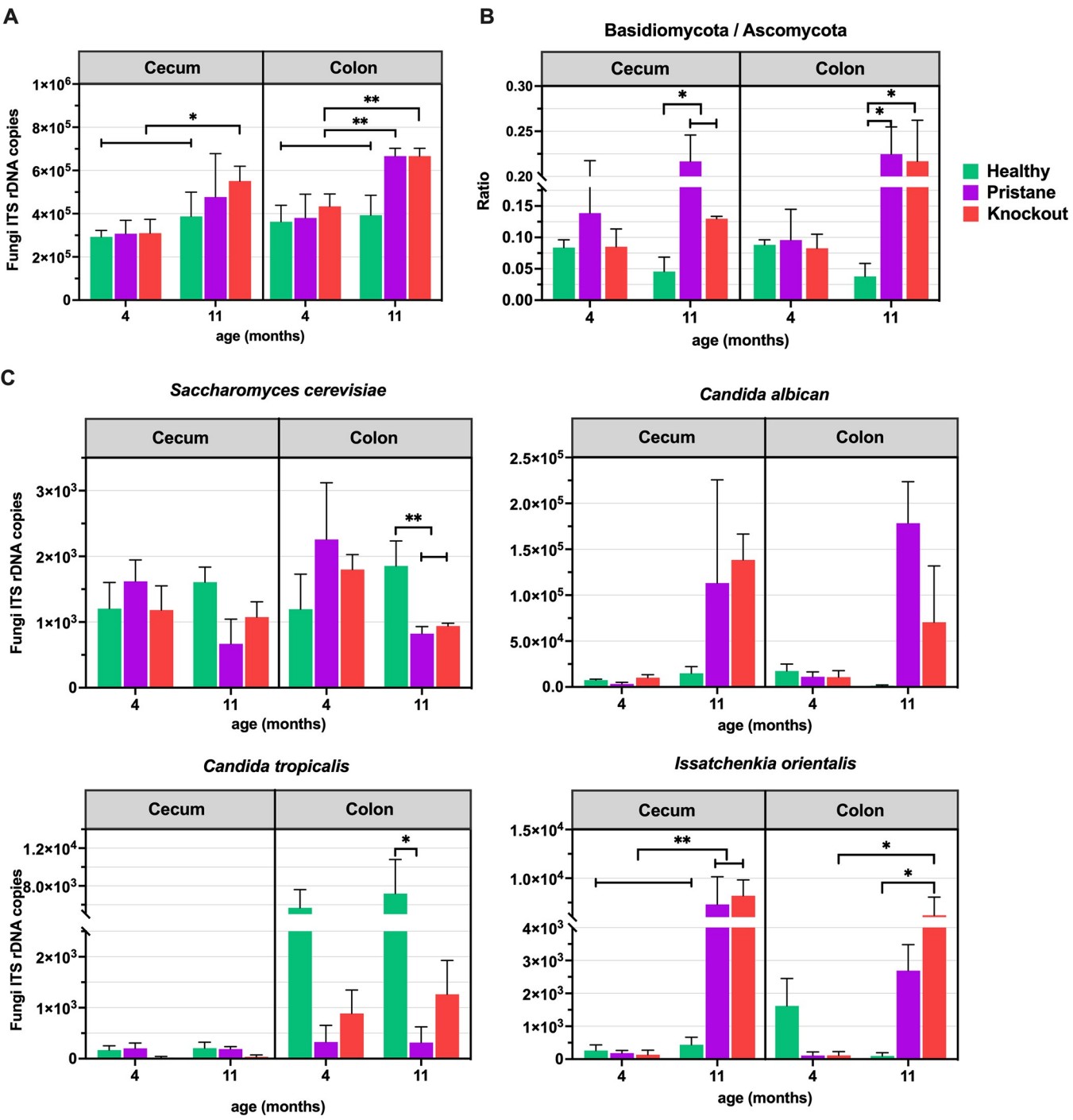

**Fig 4.** Comparison of time-dependent (young and old WT, and preclinical and established SLE by PT or KO) (A) quantitative fungal cell count, (B) Basidiomycota/Ascomycota relative abundance ratio, and (C) copy numbers of predominated statistically different species, in cecum and colon. * and ** represent ANOVA $P < 0.05$ and $P < 0.01$, respectively.

## Associations of large intestinal and fecal mycobiota with SLE mice

The correlation between gut mycobiota and lupus characteristics, including anti-dsDNA, urine protein creatinine index (UPCI), and serum cytokines (IL-6, TNF-α, and IL-10) were

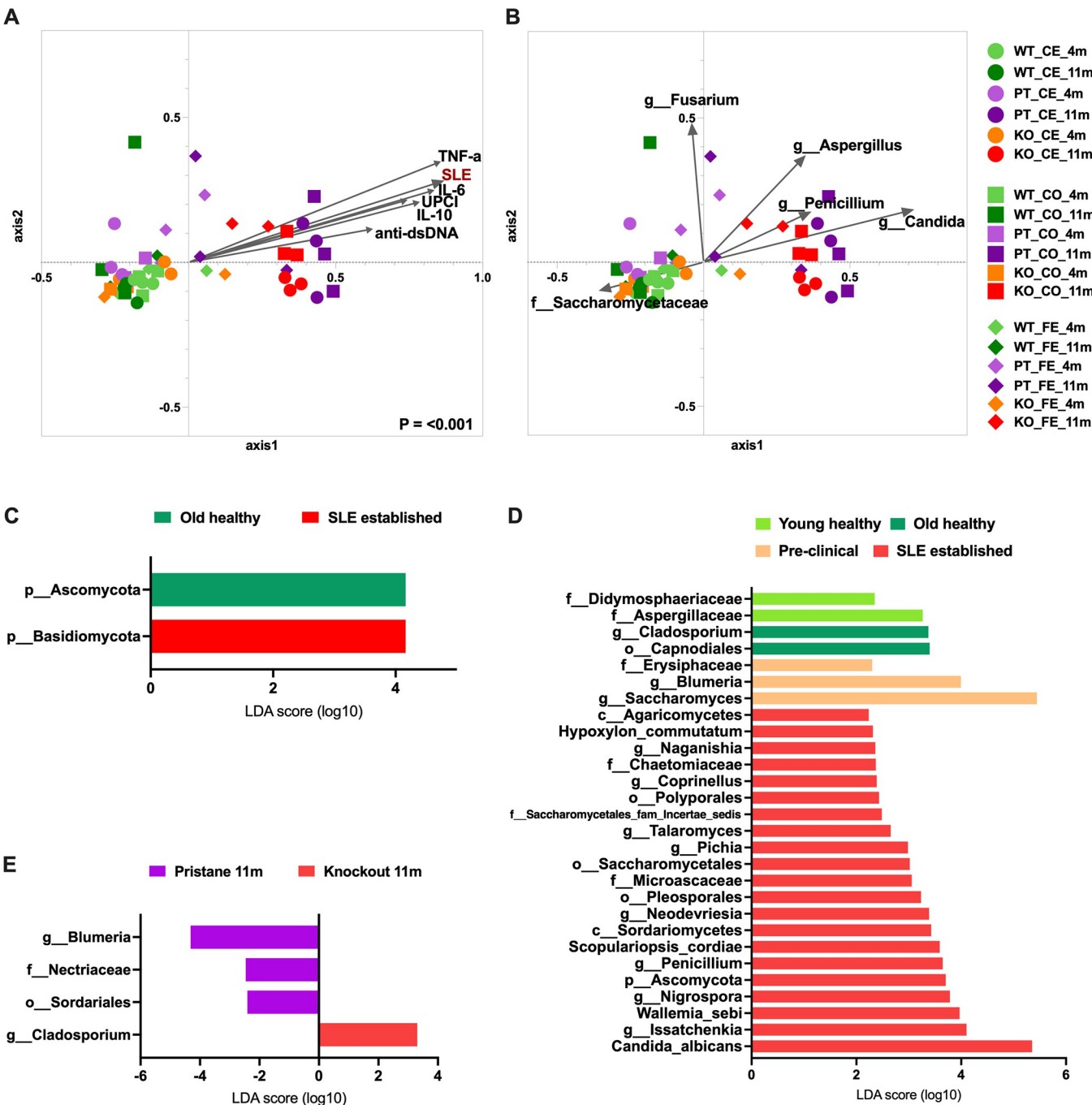

**Fig 5.** (A and B) NMDS of cecum, colon and fecal mycobiota along directional clinical parameters (SLE establishment state, anti-dsDNA antibody, urine protein creatinine index (UPCI), TNF-α, IL-6 and IL-10) and statistically representing fungal OTUs; and (C-E) linear discriminant analysis effect size (LEfSe) suggesting fungal biomarkers between different comparing groups. In (E), the SLE state PT and KO were compared. All displaying clinical parameters and fungal species here had $P < 0.05$.

examined using the NMDS with length and direction of vectors to represent strength and direction of the association. As such, the established SLE was significantly separated from the cluster of healthy and preclinical SLE, implying a positive association with lupus characteristics and gut mycobiota alteration (Fig 5A). The established SLE was influenced by *Candida* spp.

(*C. albican* and *C. tropicallis*), *Aspergillus*, *Penicillium*, and *Issatchenkia* (*I. orientails*), while *Saccharomyces* was influenced to healthy and preclinical SLE (S5 Fig and Fig 5B). Additionally, the linear discriminant analysis effect size (LEfSe) among WT (4 and 11 months), preclinical, and established SLE demonstrated elevated Ascomycota and Basidiomycota in 11 months old WT and established SLE, respectively (Fig 5C). In detailed categorizing groups, *Aspergillaceae* and *Didymosphaeriaceae* represented 4 months WT, and *Cladosporium* and *Capnodiales* represented 11 months WT, while *Erysiphaceae* and *Blumeria* (*Saccharomyces* was a conflicting between the WT and preclinical) represented preclinical SLE (Fig 5D and 5B). In parallel, there were 21 fungal species, such as *Candida albicans*, *Issatchenkia*, *Wallemia sebi*, *Nigrospora*, Ascomycota, and *Penicillium*, as a representative for established SLE (Fig 5D). For established PT and KO (11 months), *Blumeria*, *Sordariales* and *Nectriaceae* were biomarkers for PT, and *Cladosporium* was a biomarker for KO (Fig 5E).

## Discussion

Alteration in gut mycobiota alpha diversity and composition may affect immune homeostasis, bacterial community diversity, and disease characteristics [9, 10, 37, 38]. Conversely, immune responses can also affect gut microbiota [39], which may partly explain differences in microbiota composition between fecal and intestinal samples [6, 40, 41]. Therefore, this study explored mycobiota in both feces and intestinal sections through time-point analysis, focusing on preclinical lupus (under 6 months in the model) and established SLE (6 months and older in the model).

In established SLE mice, alpha diversity (variety of microbial species) showed an increasing trend in FcγRIIb-/- (KO) mice, as indicated by OTUs and Chao richness (Fig 1), suggesting an immune response impact on gut fungi. Intestinal inflammation from circulating immune complex (CIC) deposition, combined with hyperinflammatory responses in FcgRIIb-/- immune cells due to the absence of an inhibitory molecule [11, 12], selectively enhance fungicidal effect on certain fungi. This resulted in an increased Basidiomycota/Ascomycota ratio, an indicator of fungal dysbiosis seen in autoimmune diseases like IBD [10], multiple sclerosis [9], and ankylosing spondylitis [37], specifically in the large intestine of both established pristane (PT) and KO lupus models. No significant difference in gut mycobiota was found in the small intestine between established SLE mice and other groups, possibly due to short transmit time and fewer colonized fungi in this region [40].

While relative taxon abundance provides insights into mycobiota, the relative abundances of taxa are interdependent (an increase in one taxon might cause a decrease in others). Thus, absolute quantification is necessary for a comprehensive understanding of fungal dynamics within microbial community [42]. Considering the dissimilarity between fecal and large intestinal mycobiota, analyses focused on large intestine and fecal mycobiota. Notably, gut mycobiota in established SLE mice were associated with SLE characteristics, with certain fungal species serving as biomarkers. Total fungal content in feces was significantly reduced in 10-month-old SLE-prone mice, while it increased in the large intestine. This alteration in fungal content in the established lupus might be related to increased (1,3)-β-D-glucan (BG) (a major molecule component of the fungal cell wall), a major fungal cell wall component, associated with lupus-induced gut permeability defects, as previously reported in pristane and FcgRIIb-/- mice [11, 12].

The enrichment of *Candida* spp. (*C. albicans* and *I. orientalis*) in the large intestine and feces of established SLE mice might align with elevated *Candida albicans* (a common human GI tract commensal) [43] in various diseases, potentially driving gut inflammation [44]. This could be due to fungal pseudohyphal and biofilm formation, which inflame enterocytes [43].

Additionally, *Candida* spp. can support the growth of certain bacteria, impacting host health [45]. Elevated *Aspergillus* spp. in established lupus might also activate inflammation [7], while reduced *S. cerevisiae* (beneficial fungi for humans) might correlate with increased adherent-invasive bacteria [10] and reduced *Lactobacillus* spp. (beneficial bacteria) [45]. Indeed, the benefits of *S. cerevisiae* as the probiotics are well- documented in various pathologies [46, 47]. In comparison between PT and KO factors causing disease of SLE-prone mice at established SLE stage, we did not find difference in gut mycobiota composition or phylum biomarker; however, LEfSe analysis identified fungal variations, such as *Aspergillus* in PT mice and *Cladosporium* in KO mice. *Aspergillu*s and *Cladosporium* are the mold that commonly present in the gut and can affect health through *Aspergillus* toxin or volatile organic compounds from *Cladosporium* [48–51].

The causal link between fungal dysbiosis and lupus progression may involve several mechanisms. During active lupus, an altered fungal population characterized by reduced beneficial fungi (*S. cerevisiae*) and elevated pathogenic fungi (*Candida* and *Aspergillus*) that can invade enterocytes, might invade enterocytes, potentially contributing to the translocation of beta-glucan [52] from the gut into circulation, worsening systemic inflammation [53]. For instances, the overrepresentation of *Candida* species in human gut could promote intestinal inflammation, weaken gut barrier integrity and facilitate translocation of β-glucan and other fungal molecules into circulation. *Candida* species could activate dendritic cells or macrophages, leading to IL-6, IL-10, TNF-α and anti-dsDNA production, thus perpetuating autoimmunity. Systemic fungal leakage could influence the gut-kidney axis, exacerbating inflammation and proteinuria in the kidneys [17, 19–22]. Similarly, *Aspergillus* species were reported to involve gut mycobiome dysbiosis, increasing pro-inflammatory cytokines and kidney inflammation [19–22]. Thus, bacterial-fungal interactions during active lupus may be significant is possible, and exploring fungi and gut translocation of beta-glucan may yield new biomarkers or novel treatment strategies for SLE, including antifungal treatments or probiotics. This knowledge is valuable for understanding human SLE.

## Conclusions

This study presents the first evidence of time-point gut mycobiota and quantitative mycobiota from feces and the intestinal sections using the lupus models of pristane (PT) and FcgRIIb-/- (KO) mice representing lupus from environmental induction and genetic prone, respectively. The mycobiota alteration was demonstrated only in the large bowel and feces, but not in the small bowel of mice. The mycobiota of preclinical lupus (asymptomatic non-proteinuria at before 6 months old) was not different from healthy age-matched mice, while the established lupus mice (from 6 months and older) demonstrated decreased *S. cerevisiae* and increased fungi with possible pathogenicity (*Candida* and *Aspergillus*) with decreased fungal counts in feces. There was higher *Aspergillus* in PT than KO (p = 0.037) and higher *Cladosporium* in KO than PT (P = 0.0039) during the established SLE lupus stage in feces, cecum and colon. More studies on gut mycobiota alteration in patients with lupus and the therapeutic possibilities are interesting.

## Supporting information

**S1 Fig.** Rarefaction curves of quality reads per sample at phylum and genus operational taxonomic units (OTUs) of (A) all samples, (B) separate groups of healthy, pristane and KO samples.
(TIFF)

**S2 Fig. Comparison of relative copy numbers of predominated non-statistically different species OTUs between SLE mice and age-matched healthy mice at 2–10 months (*Saccharomyces*_unclassified, *Candida*_unclassified, *Blumeria* sp., *Candida nivariensis*, *Ascomycota*_unclassified, *Aspergillus*_unclassified, *Saccharomyces arboricola*, *Issatchenkia*_unclassified, *Nigrospora*_unclassified and Saccharomycetales_unclassified).** (TIFF)

**S3 Fig. Relative abundance ratio of Basidiomycota/Ascomycota in different intestinal parts at preclinical and established SLE states.** * represents statistically significant ANOVA $P < 0.05$, in comparison with healthy at 11 months. (TIFF)

**S4 Fig. Comparison of relative copy numbers of top-ten abundant genera with statistical difference between SLE (PT or KO) and WT, in cecum and colon, including *Saccharomyces*, *Candida*, *Aspergillus*, *Issatchenkia*, *Wallemia*, *Penicillium* and *Cladosporium*.** Noted that genera *Blumeria*, *Asomycota*, *Nigrospora* and *Fusarium* were not displayed as they exhibited no statistical difference. *, ** and *** represent ANOVA $P < 0.05$, $0.01$, and $0.001$, respectively. (TIFF)

**S5 Fig. NMDS of 2–10 months fecal mycobiota along directional representing fungal genera as shifting gut mycobiota composition patterns.** (TIFF)

**S1 Table.** Alpha diversity estimates at genus level of wild-type (WT), pristane (PT) and FcgRIIb-/- (KO) mice of (A) feces at 2, 4, 6, 8 and 10 months; and (B) five intestinal parts (duodenum, jejunum, ileum, cecum, and colon) at 4 and 11 months of age. (PDF)

## Acknowledgments

We acknowledged the 90th Anniversary Chulalongkorn University Fund (Ratchadaphiseksomphot Endowment Fund), Thailand Science Research and Innovation Fund Chulalongkorn University (CU_FRB65_hea(68)_131_23_61), and the Multi-Omics for Functional Products in Food, Cosmetics and Animals Research Unit, Chulalongkorn University. We thanked the National Center for Genetic Engineering and Biotechnology (BIOTEC), and the National Science and Technology Development Agency (NSTDA) for supporting us to process microbiota analyses on BIOTEC HPC server, and the Omics Sciences and Bioinformatics Center, Faculty of Science, Chulalongkorn University, for Miseq sequencing equipment and OMICS server.

## Author Contributions

**Conceptualization:** Asada Leelahavanichkul, Nattiya Hirankarn, Naraporn Somboonna.

**Data curation:** Thanya Cheibchalard, Asada Leelahavanichkul, Naraporn Somboonna.

**Formal analysis:** Thanya Cheibchalard, Naraporn Somboonna.

**Funding acquisition:** Asada Leelahavanichkul, Naraporn Somboonna.

**Investigation:** Thanya Cheibchalard, Piraya Chatthanathon, Naraporn Somboonna.

**Methodology:** Thanya Cheibchalard, Piraya Chatthanathon, Piriya Klankeo, Naraporn Somboonna.

**Project administration:** Naraporn Somboonna.

**Resources:** Asada Leelahavanichkul, Piriya Klankeo, Nattiya Hirankarn, Naraporn Somboonna.

**Supervision:** Asada Leelahavanichkul, Naraporn Somboonna.

**Validation:** Naraporn Somboonna.

**Visualization:** Thanya Cheibchalard, Naraporn Somboonna.

**Writing – original draft:** Thanya Cheibchalard, Asada Leelahavanichkul, Naraporn Somboonna.

**Writing – review & editing:** Naraporn Somboonna.

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
