## [Decision Letter · Decision Letter 0]

11 Sep 2024

PONE-D-24-20227FUNGAL MICROBIOME IN GUT OF SLE-PRONE MICE (PRISTANE AND FCGRIIb DEFICIENCY), A POSSIBLE IMPACT OF FUNGI IN LUPUSPLOS ONE

Dear Dr. Somboonna,

Thank you for submitting your manuscript to PLOS ONE. After careful consideration, we feel that it has merit but does not fully meet PLOS ONE’s publication criteria as it currently stands. Therefore, we invite you to submit a revised version of the manuscript that addresses the points raised during the review process.

It is encouraged that authors provide some insight into the differences or similarities between the 2 models and how that correlate to mycobiome to further highlight the importance of this study. Please provide point by point response to comments.

We look forward to receiving your revised manuscript.

Kind regards,

Veena Taneja

Academic Editor

PLOS ONE

Journal Requirements: When submitting your revision, we need you to address these additional requirements. 1. Please ensure that your manuscript meets PLOS ONE's style requirements, including those for file naming. The PLOS ONE style templates can be found at https://journals.plos.org/plosone/s/file?id=wjVg/PLOSOne_formatting_sample_main_body.pdf and https://journals.plos.org/plosone/s/file?id=ba62/PLOSOne_formatting_sample_title_authors_affiliations.pdf 2. To comply with PLOS ONE submissions requirements, in your Methods section, please provide additional information regarding the experiments involving animals and ensure you have included details on (1) methods of sacrifice, (2) methods of anesthesia and/or analgesia, and (3) efforts to alleviate suffering. 3. Thank you for stating the following financial disclosure: "This research was supported by the 90th Anniversary Chulalongkorn University Fund (Ratchadaphiseksomphot Endowment Fund, the Scholarship from the Graduate School of Chulalongkorn University to commemorate the 72nd Anniversary of his Majesty King Bhumibol Aduladej, and the Multi-Omics for Functional Products in Food, Cosmetics and Animals Research Unit."  Please state what role the funders took in the study.  If the funders had no role, please state: ""The funders had no role in study design, data collection and analysis, decision to publish, or preparation of the manuscript."" If this statement is not correct you must amend it as needed. Please include this amended Role of Funder statement in your cover letter; we will change the online submission form on your behalf. 4. Please amend either the title on the online submission form (via Edit Submission) or the title in the manuscript so that they are identical. 5. Please include captions for your Supporting Information files at the end of your manuscript, and update any in-text citations to match accordingly. Please see our Supporting Information guidelines for more information: http://journals.plos.org/plosone/s/supporting-information.

Additional Editor Comments:

The study is significant in defining mycobiome in 2 mouse models of lupus. However, the study lacks in defining differences between the 2 strains and how it correlates to the clinical symptoms. Do both models share or differ in a specific species which correlates to any clinical symptom in mice will be really informative. If possible, it is encouraged that authors provide those details.

Please respond to comments point by point

Reviewers' comments:

Reviewer's Responses to Questions

**Comments to the Author**

1. Is the manuscript technically sound, and do the data support the conclusions?

Reviewer #1: Yes

Reviewer #2: Yes

2. Has the statistical analysis been performed appropriately and rigorously? 

Reviewer #1: Yes

Reviewer #2: Yes

3. Have the authors made all data underlying the findings in their manuscript fully available?

Reviewer #1: Yes

Reviewer #2: Yes

4. Is the manuscript presented in an intelligible fashion and written in standard English?

Reviewer #1: No

Reviewer #2: Yes

5. Review Comments to the Author

Reviewer #1: This manuscript demonstrates changes in the gut mycobiome using two pre-clinical SLE models to identify altered mycobiota composition and diversity. Certain fungal were identified as biomarkers for different stages of SLE.

Major Points:

1. The manuscript presents the clustering status of beta diversity among the three groups in Figure 2D to define the model. However, it lacks detailed data to substantiate the definitions of preclinical SLE and established SLE in the two mouse models. Can the authors provide additional evidence or analysis to support these categorizations?

2. While the study highlights the composition of the mycobiome and its association with inflammatory factors such as anti-dsDNA, UPCI, IL-6, TNF-α, and IL-10, it does not sufficiently explore the functional implications of these observed changes. Can the authors elaborate on how these altered mycobiota affect SLE severity or activity?

3. The study does not establish a causal relationship between specific fungal and disease outcomes. It would be beneficial if the authors could incorporate experiments or data that manipulate the identified microbiota to demonstrate a direct impact on SLE progression or symptoms.

4. The English needs improvement for better readability.

Reviewer #2: The manuscript addresses the differentiation of fungal microbiota between systemic lupus erythematosus (SLE) model mice and healthy controls. The authors demonstrated that the microbiota in two SLE model mice, the pristane-induced mouse model and the Fc gamma receptor RⅡB-deficient mouse model, representing environmental and genetic factors respectively, showed increased fungal diversity, elevated specific gut fungal species, and an altered Basidiomycota/Ascomycota ratio in correlation with lupus characteristics. Correlations between gut microbiota and autoimmune diseases such as rheumatoid arthritis have been reported. The authors' research is interesting and contributes valuable knowledge toward understanding the role of gut microbiota in the pathogenesis of SLE. However, the labels in the figures of this manuscript are difficult to read. Therefore, I recommend acceptance of this manuscript after the figures are amended for improved clarity.

6. PLOS authors have the option to publish the peer review history of their article (what does this mean?). If published, this will include your full peer review and any attached files.

Reviewer #1: No

Reviewer #2: **Yes: **Hiroki Akazawa

---

## [Author Response · Author response to Decision Letter 0]

14 Oct 2024

Dear Reviewers,

Thank you for valuable and generous reviews. We revised, corrected, and uploaded the manuscript (clean), the revised manuscript with track changes (using track changes), Figures, and supplements, following the comments and advices. Line numbers are based on the "Manuscript w/ Track Changes (Simple Markup)" file. 

Thank you very much.

Sincerely yours, 

Naraporn Somboonna (corresponding author)

Asada Leelahavanichkul (co-corresponding author)

---

## [Decision Letter · Decision Letter 1]

8 Nov 2024

PONE-D-24-20227R1Fungal microbiome in gut of systemic lupus erythematosus (SLE)-prone mice (pristane and FCGRIIb deficiency), a possible impact of fungi in lupusPLOS ONE

Dear Dr. Somboonna,

Thank you for submitting your manuscript to PLOS ONE. After careful consideration, we feel that it has merit but does not fully meet PLOS ONE’s publication criteria as it currently stands. Therefore, we invite you to submit a revised version of the manuscript that addresses the points raised during the review process.

**ACADEMIC EDITOR: **

**The manuscript has certain statements that need clarification. To make it suitable for all readers, please make sure to read it for proper English so it is easily understood. **

We look forward to receiving your revised manuscript.

Kind regards,

Veena Taneja

Academic Editor

PLOS ONE

Journal Requirements:

Additional Editor Comments:

Please revise the article for proper English.

Line 145--- :At complete study, mice----"

Line 159 " what does it mean " All mice were measured SLE characteristics"

Clarify these statements and proof for other mistakes.

Reviewers' comments:

Reviewer's Responses to Questions

**Comments to the Author**

1. If the authors have adequately addressed your comments raised in a previous round of review and you feel that this manuscript is now acceptable for publication, you may indicate that here to bypass the “Comments to the Author” section, enter your conflict of interest statement in the “Confidential to Editor” section, and submit your "Accept" recommendation.

Reviewer #1: (No Response)

Reviewer #3: All comments have been addressed

2. Is the manuscript technically sound, and do the data support the conclusions?

Reviewer #1: (No Response)

Reviewer #3: Yes

3. Has the statistical analysis been performed appropriately and rigorously? 

Reviewer #1: (No Response)

Reviewer #3: N/A

4. Have the authors made all data underlying the findings in their manuscript fully available?

Reviewer #1: (No Response)

Reviewer #3: Yes

5. Is the manuscript presented in an intelligible fashion and written in standard English?

Reviewer #1: (No Response)

Reviewer #3: Yes

6. Review Comments to the Author

Reviewer #1: (No Response)

Reviewer #3: Though this study is not anew subject, it has great impact on understanding how autoimmune diseases operate and has impact in translational medicine and inventing new immunological treatments

7. PLOS authors have the option to publish the peer review history of their article (what does this mean?). If published, this will include your full peer review and any attached files.

Reviewer #1: No

Reviewer #3: **Yes: **Becky Abdissa Adugna

---

## [Author Response · Author response to Decision Letter 1]

10 Nov 2024

Decision: Revision required [PONE-D-24-20227R1] - [EMID:2e0c38865e667e61]

Dear editor and reviewers,

We revised and corrected the manuscripts (using track changes) following the comments. Line numbers are based on the "Manuscript w/ Track Changes (Simple Markup)" file. 

Thank you very much.

Sincerely yours, 

Naraporn Somboonna (corresponding author)

Asada Leelahavanichkul (co-corresponding author)

---

## [Editor Report · Decision Letter 2]

12 Nov 2024

PONE-D-24-20227R2Fungal microbiome in gut of systemic lupus erythematosus (SLE)-prone mice (pristane and FCGRIIb deficiency), a possible impact of fungi in lupusPLOS ONE

Dear Dr. Somboonna,

Thank you for submitting your manuscript to PLOS ONE. After careful consideration, we feel that it has merit but does not fully meet PLOS ONE’s publication criteria as it currently stands. Therefore, we invite you to submit a revised version of the manuscript that addresses the points raised during the review process.

We look forward to receiving your revised manuscript.

Kind regards,

Veena Taneja

Academic Editor

PLOS ONE

Journal Requirements:

**Additional Editor Comments:**

Revised sentences are not grammatically correct.

Please get the manuscript checked for English.

---

## [Author Response · Author response to Decision Letter 2]

12 Nov 2024

Decision:: Revision required [PONE-D-24-20227R2] - [EMID:45c95f34cb950ce7]

Dear Editor and Reviewers,

We revised following the comments and advice. 

Thank you very much.

Sincerely yours, 

Naraporn Somboonna (corresponding author)

Asada Leelahavanichkul (co-corresponding author)

---

## [Editor Report · Decision Letter 3]

14 Nov 2024

Fungal microbiome in gut of systemic lupus erythematosus (SLE)-prone mice (pristane and FCGRIIb deficiency), a possible impact of fungi in lupus

PONE-D-24-20227R3

Dear Dr.Somboonna

We’re pleased to inform you that your manuscript has been judged scientifically suitable for publication and will be formally accepted for publication once it meets all outstanding technical requirements.

Kind regards,

Veena Taneja

Academic Editor

PLOS ONE

Additional Editor Comments

The manuscript has improved. However, still some changes are required.

Line 185 , please change it to "All mice were used to measure ------ to confirm SLE". Serum measurements are not clinical characteristics.

Line 145, please correct it - "At the completion of the study-----
---

## [Editor Report · Acceptance letter]

26 Nov 2024

PONE-D-24-20227R3 

PLOS ONE

Dear Dr. Somboonna, 

I'm pleased to inform you that your manuscript has been deemed suitable for publication in PLOS ONE. Congratulations! Your manuscript is now being handed over to our production team.

Kind regards, 

on behalf of

Dr. Veena Taneja 

Academic Editor

PLOS ONE